# ORCHESTRATING TOOL ECOSYSTEM OF DRUG DISCOVERY WITH INTENTION-AWARE LLM AGENTS

## ABSTRACT

Fragmented tools and models and complex decision-making with incomplete and heterogeneous information often hinder the drug discovery process. Large Language Models offer promising capabilities in commonsense reasoning and tool integration, yet their application in drug discovery remains constrained by challenges such as being incapable of handling large tool space, limited planning capabilities based on scientific intentions, and unscalable evaluation. We introduce GENIEAGENT, a drug discovery agent that integrates a wide range of molecule design models and bridges the user intentions to concrete actions by navigating the large skill ecosystem. By unifying disparate tools under a single natural language interface, GENIEAGENT enables cross-tool reasoning and supports complex scientific workflows. We also propose an evaluation framework simulating drug discovery conversations, based on real-world experiments. A large-scale assessment, validated by expert annotations, demonstrates that GENIEAGENT reliably meets the majority of molecular engineers' needs with high scientific accuracy and robustness.

## 1 INTRODUCTION

The early development stage of a drug highly depends on the models and tools that are used to measure the properties of molecules, rank candidate molecules, and generate revised or brand-new sequence designs with desired properties. These capabilities encompass both non-differentiable operations, such as bioinformatics tools, and differentiable models, including fine-tuned machine learning networks. However, these tools are often developed independently, trained on different datasets, and implemented using diverse architectures (Liu et al., 2023; McNaughton et al., 2024). This fragmentation disrupts the drug discovery workflow, slows down the feedback loop, and creates barriers to tool accessibility and usability (Tu et al., 2024). Beyond the challenge of integrating molecular design tools, drug discovery is inherently an open-ended exploration that demands careful reasoning and planning, requiring scientists to compose individual actions and tools to effectively address complex scientific objectives.

Large Language Models (LLMs) have been shown to perform well for commonsense reasoning, natural language understanding and tool using (Swan et al., 2023; Rajendran et al., 2024). Though many works have been utilizing LLM agents to connect tools for scientific discovery (Abbasian et al., 2024; Li et al., 2024; Ferruz & Höcker, 2022; Huang et al., 2024), several bottlenecks exist that prevent broad adoption of the LLM agents in scientific discovery processes. Firstly, existing works support only a few tools, without the feasibility and robustness of navigating an ecosystem with a large amount of expert-curated tools. Secondly, the orchestration of those tools follows an expert-defined order, preventing complex tool planning and interaction from vague user intention. Additionally, the drug discovery process is open-ended, and domain expertise is required to use and evaluate the system; there is a lack of efficient approaches to evaluate multi-turn scientific discovery agents.

In this work, we propose GENIEAGENT, a drug discovery agent connecting to a large-scale domain-specific tool ecosystem with scientific intention awareness. We first curate a collection of drug discovery models and tools, enabling large-scale molecule scoring, ranking, and generative capabilities. These tools cover a wide spectrum of molecule design steps, involving both large and small molecule spaces. Various types of models are incorporated, *e.g.*, generative, scoring, searching, to

address a wide range of design objectives such as hit expansion, lead optimization, compound filtering and ranking. The comprehensive suite of capabilities enables us to navigate an unprecedented action space for drug discovery agents.

We then propose novel agent design innovations to tackle the challenges of the scientific discovery agents mentioned above. To map the high-level and ambiguous scientific intention to actionable tool uses, we introduce a synthesized intention index that provides reference intention and solutions to facilitate the reasoning and planning of the agent. We design mechanisms to enable the navigation of a large collection of tools with specialized skill-specific agents and metadata-inspired searching tools. We finally introduce the *hint routing nodes*, a new paradigm of providing routing guidance to the agent by appending pseudo reminder messages to the memory. Hint nodes combine the benefits of fixed workflow and open-ended exploration. This lightweight approach guides the agent with critical actions and plans in mind, preventing hallucination while keeping flexibility. These efforts unify the separate drug discovery tools under a single natural language interface, enabling cross-tool reasoning and orchestrating a scientific workflow with multiple capabilities.

To evaluate GENIEAGENT, we also propose and perform a large-scale evaluation mechanism that simulates the drug discovery conversation based on real-world drug discovery experiments. We create test cases consisting of scientific intentions, model selections, prepared data, and model configurations induced from real-world scientific research logs. We then propose an evaluation agent bounded with information leaking tools that gradually provide more complete and clear goals and data, simulating a vague-to-concrete scientific exploration process. This evaluation framework enables us to do large-scale scientific discovery agent evaluation and ablation studies.

We perform quality assessments with both automatic pipelines and expert ratings based on scientist-in-the-loop conversations with chemists and molecular biologists who perform real-world drug discovery campaigns. The result indicates that GENIEAGENT can deliver the majority of the needs of molecular biologists or medicinal chemists with high reliability and robustness in terms of scientific factuality. Compared with existing agent designs like ReAct, the unique design of GENIEAGENT demonstrates significant superiority for overall success rate and turn-level quality.

## 2 RELATED WORKS

Existing works explore using LLM agents for scientific discovery (Gao et al., 2024). Some works frame the scientific discovery tasks in a closed environment with verifiable outcomes, such as code generation for scientific problems Laurent et al. (2024); Swan et al. (2023); Romera-Paredes et al. (2024) or conducting research in a virtual simulated environment Jansen et al. (2024). Some works focus on training LMs to directly equip them with scientific reasoning and action capabilities, *e.g.* manipulating protein sequence or changing protein properties (Ma et al., 2024). Boiko et al. (2023) incorporate tools for Google searching, Python code execution, searching documentation and calling experiment API for autonomous chemical research. McNaughton et al. (2024); Kang & Kim integrate a number of tools to an agent with a simple framework such as ReAct (Yao et al., 2023). Similar designs are used for various scientific tasks, such as catalyst design Sprueill et al. (2023), gene-editing experiments Huang et al. (2024), genomics question answering Jin et al. (2024), and material design Kang & Kim (2024). Ye et al. (2023) rely on a single LLM to do all tasks while specific training data and architectures are needed for different tasks, limited by its poor performance and scalability. The simplicity of the agent design, the limited integrated tools' scope, and the lack of iterative dynamic conversation capabilities make the adaptation and application of existing scientific discovery agents particularly challenging.

## 3 DRUG DISCOVERY CAPABILITY ECOSYSTEM

### 3.1 SPECIALIZED DRUG DISCOVERY MODELS

Antibody design and small molecule design are both crucial yet highly challenging aspects of drug discovery. Antibodies offer high specificity and affinity, but designing them requires balancing stability, manufacturability, and immune evasion (Frey et al., 2023). Small molecule design involves navigating vast chemical space to find compounds with optimal pharmacokinetics, target specificity, and minimal off-target effects Pinheiro et al. (2023). Both processes involve multiple constraints, making them ideal for AI-driven approaches like GENIEAGENT, which can efficiently explore molecular spaces, predict interactions, and suggest novel candidates.

GENIEAGENT orchestrates a wide range of drug discovery models that span the entire drug design pipeline for both antibodies and small molecules, from sequence design to property optimization. We lay out model details and input-output specifications in Table 4.

**Generative models proposing candidate molecules.** The method suite includes generative models that propose candidate molecules, such as an antibody design method with implicit guidance (Tagasovska et al. (2024), Table 4, row 1), affinity-guided antibody maturation (Gruver et al. (2023b), row 2), and a latent 3D generative approach for small molecule design (Nowara et al. (2024), row 3).

**Property prediction models.** These generative tools are complemented by multiple property prediction models, including antibody developability assessment using molecular surface descriptors (Park & Izadi (2024), row 4), antibody expression and antibody-antigen complex prediction (Gruver et al. (2023a), row 5), and antibody profiling based on hydrophobicity and charge descriptors (Raybould & Deane (2022), row 6).

**Structural analysis methods.** Additional structural analysis methods provide insights into antibody properties, including ABangle for orientation characterization (Dunbar et al. (2013), row 7), PEP-Patch for electrostatic surface patch estimation (Hoerschinger et al. (2023), row 8), and spatial aggregation propensity scoring for identifying aggregation-prone regions (Waibl et al. (2022), row 9). We include multiple tools for molecular docking and scoring. Protein structures are prepared using SPRUCE (Baell & Holloway (2010), row 10), ensuring compatibility with docking pipelines. Ligand docking is performed using POSIT (row 11) and HYBRID (row 12), which generate ligand poses within binding pockets(Baell & Holloway, 2010). Docking poses are then evaluated using HYBRID scoring (row 13), as well as GNINA (rows 14 and 15), which integrates deep learning-based pose scoring with traditional docking methods (McNutt et al., 2021). Finally, metabolite prediction models (row 16) assist in evaluating small molecule modifications, identifying metabolic transformations of drug candidates and their corresponding probabilities(Coley et al., 2017; Djoumbou-Feunang et al., 2019).

Together, this diverse set of tools provides a comprehensive suite for generative design, molecular property assessment, and docking-based screening in drug discovery.

## 3.2 AUXILIARY AND GENERAL TOOLS

We introduce additional tools to access enhanced scientific knowledge and support personalized user queries. We also include a Python code interpreter tool to execute Python scripts to facilitate ad-hoc calculation, data processing of the provided file, presenting aggregated results and open-ended data analysis.

**Scientific search tools.** We build semantic indexes for PubMed and ScienceDirect and develop a search tool to retrieve the relevant context and evidence for the user query. Another tool connects to DuckDuckGo and provides web search results to access broad, up-to-date information. All these search tools return a list of relevant paragraphs.

**Personalized execution retrieval tools.** GENIEAGENT also supports personalized scientific discovery by allowing users to query and reason based on their previous experiments. Scientists' actions and experiments are saved for future queries. To enable the scientists to interact with their historic experiments and previous efforts, we introduce a user log retrieval tool to query the database and retrieve recent experiment logs related to a specified model or around a specific time. The function would return the records of the matched experiments, including input, produced results, and metadata.

## 4 GENIEAGENT DESIGN

GENIEAGENT is designed to assist drug discovery scientists in progressively conducting experimentation, predicting, and processing actions through a multi-turn conversation between users and the agent. The conversation starts with a high-level scientific intention from the user and ends after appropriate actions are conducted. Different from the agents that take initial instructions and then autonomously act, our design puts scientists in the loop. The agent is expected to respond and adapt to additional information, requests, and instructions provided during each user's turn.

We introduce the overall architecture of GENIEAGENT in section 4.1. We then introduce three aspects of novel techniques to address the key challenges of large-scale scientific discovery agent

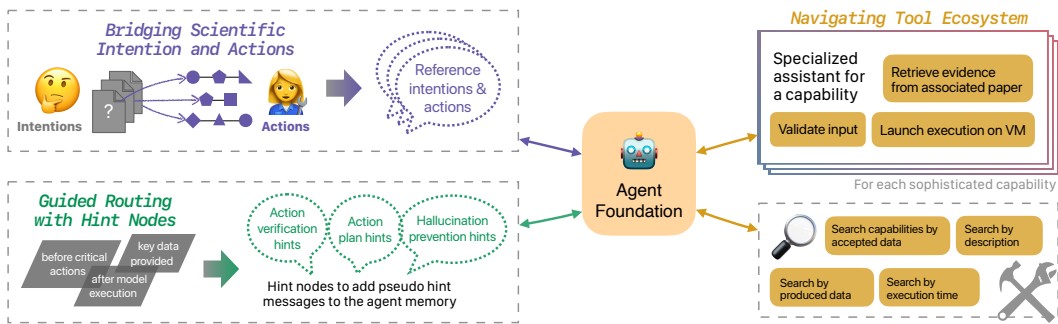

Figure 1: Agent design of GENIEAGENT. The agent design uses a synthesized intention-action pool to inform the agent of possible trajectories to bridge scientists' high-level intention with concrete steps. GENIEAGENT uses index-inspired searching tools and specialized agents for each capability to make sure the agent scales and can handle large action space. Finally, it uses hint nodes to add timely reminders to the memory to guide the agent routing.

design. First, the significant gap between the users' high-level intention and concrete tool-calling actions challenges the reasoning and planning capabilities of the agent. We bridge this gap by retrieving reference intention-action pairs shown in section 4.2. Second, the large number of supported tools makes it difficult for LLMs with existing agent design, *e.g.* ReAct, to navigate the ample action space. The specialized assistant design and indexing tools introduced in section 4.3 address the challenge of the large action space. Finally, we use hint nodes as part of the agent routing graph described in section 4.4, balancing guidance and flexibility to keep the conversation on track toward the goal and prevent hallucination.

## 4.1 OVERALL ARCHITECTURE

GENIEAGENT is built on top of an LLM with an assigned system role (shown in Appendix A.2.1) as the primary agent and a memory that keeps the conversation histories. A state dictionary is maintained to explicitly track the *action plan* (*e.g.* prepare protein structures with SPRUCE, then generate ligand poses within binding pockets using POSIT) and *values* (*e.g.* heavy and light chains, antigen sequences) to be used as potential input for the models. The action space state parameters help the agent keep progressing for multi-step actions, and the values make the essential input unchanged and easily recalled after long conversations.

We design a supervisor agent architecture with shared memory among agent profiles. The *primary agent* has access to the auxiliary and general tools introduced in section 3.2. The primary agent focuses on planning actions according to the user's intention and assists in exploring agent capabilities. For each sophisticated model in section 3.1, the *specialized agent* is created to handle model-specific queries. We further describe the interactions between two kinds of agents in section 4.3.

## 4.2 BRIDGING SCIENTIFIC INTENTION AND ACTIONS

The expanding nature, large option pool and fine-grained difference of the supporting capabilities in the tool ecosystem make it unrealistic for users to be aware of potential concrete actions. The users' utterances, especially in the early stage of the conversation, would mostly be about their high-level scientific intentions without mentioning specific models to be used. The agent is required to create action plans according to the intentions by understanding the potential sub-steps, requesting clarification and additional input from users, and navigating skill information. We synthesize a pool of initial intentions and corresponding action chains and use them as references for the planning agent to bridge the intention-to-action gap.

**Constructing intention-action pairs.** There is no existing data that includes drug discovery intentions and corresponding steps to address them. To obtain a reasonable size of intention-action pairs without expensive expert annotation, we propose a self-play agent to produce both intention and action chains. According to the input and output specification defined as parts of the tools, we first curate a set of valid action chains where the upstream model's output data type overlaps with the downstream model's input. We then reversely generate potential intention that leads to an action chain with a self-play agent based on GPT-4o. We provide the actions with related descriptions to

enrich the context of the considered steps in the action chains, in addition to a few similar action chains as negative examples to guide the LLM in generating an intention that only applies to the target positive action chain.

**Referencing similar intention and actions.** When the primary agent responds to each turn, we find the top intentions in the reference pool that semantically match the user query and then append the selected intentions and their action chains as part of the conversation history. These references are added before any tool-calling and reasoning of the primary agent so that the references benefit all primary agent operations. Note that the reference retrieval module is not used as a tool, optionally called by the primary agent, which would limit its effective scope. We also do not use the retrieval results as in-context examples as part of the agent query since the query can fall into a wide range of topics and may not directly benefit from the references.

## 4.3 Navigating Large Action Space of Drug Discovery Capabilities

The size of the capability ecosystem is large, and many models can be hard to distinguish without domain expertise or understanding the model details. Configurations of all capabilities might not even fit in the context of some base LMs. Instead of binding all tools directly to the LLM, we use searching utility tools inspired by different indexing of the capabilities to locate the appropriate concrete models. We offload model-specific tools, such as model-specific QA, receiving and validating input data and launching execution, to specialized agents.

**Metadata-indexed searching tools.** We created multiple searching tools that return the appropriate list of recommended capabilities given a query of model description, required input, or expected output data formats. These utility tools are part of the primary agent to facilitate the action planning stage. These categorizations based on different organizational criteria match the potential source of a drug discovery initiative. When the scientist has a certain kind of data in hand, searching with acceptable input can be recommended as the first step. For the tasks with a firm expectation of certain types of results, categorizing capabilities by output would be called.

**Specialized agents for capability-focused tasks.** The supervising specialized agent design separates the execution process from the planning phase done by the primary agent. The supervisor agent separation design enables the scalability of GENIEAGENT to accommodate any number of drug discovery capabilities. When new capabilities are added, there is no additional context window taken for the primary agent as the model information is obtained through the metadata-indexed searching tools. When responding to queries of a specific model (*e.g.*, asking about the training data being used and the evaluation performance of the model) or executing a specific model, these capability-specific tasks are done by the corresponding specialized agents without distracting and potentially noisy information about other capabilities.

Sophisticated drug discovery capabilities, like the ML models, are complicated actions involving acquiring, processing, and validating the input, data loading, and asynchronous execution in virtual machines. A specialized agent is created for each sophisticated action with a separate system role and access to the model-specific tools. The specialized agent is instructed to prompt the user to provide the required input data, validate the input with a validation tool, and confirm the filled data is correct. After receiving the confirmation, the specialized agent calls a launch tool to launch a script on a virtual machine with the provided data.

After the primary assistant has confirmed the action plan, the execution is done by specialized assistants for corresponding skills. The primary agent can choose to route to one of the specialized assistants once an action plan is created. A proxy tool node for each specialized assistant is created and bonded to the primary agent, where each tool calling would route the agent flow to the mapped specialized assistant. When the current state is in a specialized agent, the router can choose to jump out of it and route back to the primary agent if the specialized one detects that the user's query is beyond the focused scope.

## 4.4 Guided Routing with Hint Nodes

**Hint nodes to balance controlled flow and flexibility.** Dynamic instructions to the agent emerge on the fly depending on the outcome of upstream conversations. Including all instructions in the static system role is not feasible. On the other hand, defining a fixed routing graph limits the generalizability and flexibility of the agent. Thus, we introduce the hint routing nodes to guide the agent with dynamic instruction by appending system turns to the conversation history with a reminder message. The hint node is implemented as a routing node for the agent, if the certain condition is

met, then the only operation of this node is to add a hint conversation turn. This technique enables us to guide the agent with the following functions while keeping flexibility enabled by the strong reasoning skills of the underlying LLMs.

**Verifying critical actions.** We require the agent to confirm that the user is satisfied with the produced action plan before entering specialized assistants to prepare the input and execute it. Additionally, before launching the execution jobs given user-provided data, we require the agent to confirm the inputs are correct from the user. To achieve both confirmations, before entering specialized assistants or execution, hint nodes are added to remind the agent to receive confirmation from the users. The hind message is provided in Appendix A.2.3.

**Following action plan.** Since each skill's execution can potentially require dozens of turns to receive input, validate and launch the job, the primary agent could easily lose track of the remaining actions in the plan. Thus, a hint node reminding the agent of the saved plan in the state is added after routing back to the primary agent from a specialized one. The action plan is saved as a natural language sentence right before entering any specialized agents. In the hint message, we retrieve the plan from the tracked state and include it as part of the pseudo-utterance, as further elaborated in Appendix A.2.4.

**Handling hallucinated values.** After receiving user input, such as heavy and light chain sequences, those values are saved to the state memory of the agent. If the input to a drug discovery ML model is not part of the user query, it could be hallucinated by the LLM if the user does not explicitly ask the LLM to generate candidate values from scratch. In that case, a hallucination handling hint node is added before executing the job to remind the agent to confirm the source of input values and potentially correct values with unknown sources.

## 5 AUTOMATIC AGENT EVALUATION WITH MULTI-AGENT CONVERSATION SIMULATION

To achieve a scalable and fine-grained agent conversation assessment for the drug discovery domain, we need to have ground-truth results and a mechanism to handle multi-turn conversations. Existing works either use human annotators to provide such ground truth or rewrite existing test instances for enhanced diversity (Zhu et al.). However, there is no such dataset with drug discovery scientists' intentions paired with experiments. When handling multi-turn conversations, most of the existing works do not provide a fine-grained turn-wise evaluation. Some works simulate a close environment (Zhou et al., 2024), such as web and OS (Xie et al., 2024), or match the generated conversation with reference multi-turn dialogue (Liu et al., 2024). However, these methods are infeasible to extend to open-ended drug discovery tasks.

We propose a novel evaluation framework for the open-ended drug discovery setting consisting of 1) test case creation inspired by real-world drug discovery efforts illustrated in section 5.1, 2) multi-agent high-quality scientific discovery conversation simulation that mimics the scientific thinking processes shown in section 5.2, and 3) automatic scoring for outcome and process quality evaluation introduced in section 5.3. In this section, we introduce the evaluation setup. We then report the automatic evaluation results in section 6 and additionally provide human evaluation results in section 7. This automatic evaluation framework enables scalable quality assessment of the agent design.

The end-to-end agent evaluation starts with an initial intention that the scientists would like to achieve with the agent, carries out multi-turn conversation to concertize the action plan and explicitly observe the detailed intention, and ultimately takes the actions to perform corresponding experiments.

### 5.1 TEST CASES CREATION INSPIRED BY REAL EXPERIMENTS

We create a set of test cases consisting of three items: 1) the initial scientific intentions, 2) the prepared data (antibody sequence or PDB files) and configurations (such as model hyperparameters), and 3) corresponding concrete model selection actions (such as one or more capabilities listed in Table 4) based on the specified capability ecosystem. The prepared data and model selections are based on experiment logs in real-world drug discovery efforts. However, the intentions that lead to those actions are not recorded and would be expensive to annotate due to the expert cost. We propose to generate those intentions as silver-labeled starting points for each conversation.

When scientists approach GENIEAGENT, they are mostly not even clear about their intention and finalize the data to be used. To simulate the scientific thinking process and make the test cases more

realistic, we need several versions of the intentions and data. These versions should include a clearer and more concrete one with all the details possible, along with some versions with less information, incorrect format and misleading instructions.

**Real-world drug discovery experiments.** We collect 343 drug discovery experiments performed by scientists using the same drug discovery capability ecosystem introduced in section 3. These jobs are launched by real biologists and bioinformatics scientists for real-world drug development. 54% of these experiments focus on taking the heavy and light chains as inputs, 14.7% of these efforts take a PDB file as input, and 11% of them work on SMILES sequence.

**Iterative vague intention generation.** Given the input and configurations of these experiments, the intentions and goals of the scientists while these experiments are launched are not recorded. We conduct an iterative process to reduce the information and details from the complete input and model selection judgment to produce several versions of compromised and vague user intention. We prompt a GPT-4o model to generate a summarized potential intention of the scientist. The input prompt includes the model selection, model description and the type of the model (ranking, scoring or generation). Given all these inputs, we generate two variants, a 1-2 sentence one and a 5-10 word one. Based on these two variants, we iterate the generation again by only providing the generated abstract intentions and prompt the GPT-4o model to summarize the two intentions to be more abstract and vague, producing another two variants of the user intentions.

**Compromised input generation.** User inputs, such as heavy chains, light chains, and SMILES sequences, are what the scientists expect the agent to launch experiments on. These input arguments are not prepared ahead of time, and the scientist might change their mind during the conversation with the risk that the agent could hallucinate random input. To simulate the process of concertizing the exact input, we iteratively generate several compromised versions reversely from the ground-truth input and configurations. We use heuristic functions to compromise the input by removing an input argument entirely, producing a shorter version, or replacing it with a similar but ungrounded input generated by GPT-4o without any evidence.

With these techniques, we obtain four user intention variants and three user input variants. In total, 343 test cases with compromised intention and input are generated for conversation simulation.

## 5.2 MULTI-AGENT EXPERT CONVERSATION SIMULATION

We construct an evaluation agent acting as a drug discovery scientist to chat with GENIEAGENT as illustrated in figure 2. The evaluation agent is able to simulate the scientific discovery process with the aid of an agent due to 1) specialized system role and 2) iterative detail exposure with tool use. The evaluation agent is based on GPT-4o with two tools bound. It acts based on a system role instruction that contains the most abstract intention and most compromised input, both generated in section 5.1. We provide a tool that could return a more concrete user intention, and another tool that could return a more complete user input and configurations, for the evaluation agent. The evaluation agent is instructed to call these tools if it decides that more information or

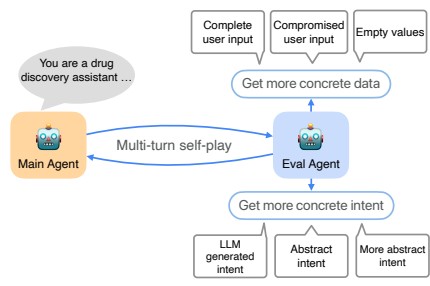

Figure 2: Multi-agent expert conversation simulation.

clarification is needed. During the conversation, the evaluation agent treats the response from GENIEAGENT as the user utterance, simulating the scientific discovery process where the scientist becomes more aware of their goals and finalizes their input choices. The simulated conversation ends when GENIEAGENT finishes the execution of all planned actions or the evaluation agent produces the ending signal, which is part of its system instruction. The results of the simulation would be a multi-turn conversation and the ultimate capability selections made by the evaluating target agent GENIEAGENT.

## 5.3 OVERALL AND TURN-LEVEL METRICS AND LLM-BASED SCORING

We evaluate the agent's capabilities with metrics reflecting both ultimate and intermediate results annotated by experienced drug discovery experts. For the end-to-end evaluation, **overall success rate** is calculated to reflect the percentage of successful execution of the correct input arguments and configuration parameters. To be successful, three conditions have to be met: 1) the agent chooses

the same chain of capabilities as the ground-truth action chains in the test cases, 2) the input data to all models when executing those models is correct, and 3) the configuration input to all models must match the gold labels. We additionally report the **model selection success rate** to reflect the percentage that the agent chooses the correct model(s) to meet users' needs. The overall success is a stricter criterion than the model selection success.

For turn-level intermediate performance, we annotate the quality of each turn according to the following dimensions. We then average ratings of each dimension across all turns from various simulated conversations to produce the final turn-level scores. These dimensions include: 1) **Factuality**: whether the output from the agent is free from scientific errors; 2) **Progressiveness**: whether the output helps to make progress toward the ultimate goal of launching the correct experiment; 3) **Informativeness**: whether the output makes the user more clear about what happens and what will happen without confusion about the agent's actions.

We use an LLM as the judge for each quality dimension with separate system role profiles. An expert-curated system role includes the definition of each metric, the comprehensive information from the test cases (*i.e.*, ultimate model selection, complete input data, full configurations for the selected models), and the instruction to rate the quality of each turn from 1-5. Previous turns are also provided for better judgment of the target turn.

# 6 AUTOMATIC EVALUATION RESULT

## 6.1 EVALUATION SETUP

**Baselines.** We compare the performance of GENIEAGENT with two baselines supporting multi-turn conversation. **LLM with Tools** is a simple agent based on GPT-4o with access to the tools that could directly launch the supporting models described in section 3. The descriptions and IO specifications of all capabilities are passed to the LM's context. **ReAct with Tools** is an agent with the same design as the first baseline but using a ReAct agent framework. Both comparing designs use the same system role as GENIEAGENT.

## 6.2 RESULTS FROM AUTOMATIC EVALUATION ON SIMULATED DRUG DISCOVERY CONVERSATIONS

Table 1: Drug discovery model orchestration performance for simulated conversation generated by the multi-agent evaluation framework. We provide the tools that could accept input and invoke the corresponding models for the two baselines. Overall scores are calculated by matching the ground-truth input and model selection of the test cases. The turn-level ratings are averaged across LLM-judged annotation for each turn's quality in terms of actuality, progressiveness, and informativeness. We use GPT-4o for all experiments.

| | Method | Overall (0-100%) | | Turn-level (Averaged 1-5 ratings) | | |
|---|---|---|---|---|---|---|
| | | Overall SR | Model Selection SR | Factuality | Progressiveness | Informativeness |
| 1 | LLM with Tools | 12 | 24 | 3.4 | 2.7 | 3.2 |
| 2 | ReAct with Tools | 14 | 34 | 3.1 | 3.1 | 3.5 |
| 3 | GENIEAGENT | 64 | 72 | 4.8 | 4.6 | 4.8 |

Table 1 presents the performance comparison for drug discovery model orchestration capabilities. We observe that GENIEAGENT can achieve the user's intended goals in most cases with an overall success rate of 64% and a model selection success rate of 72%. GENIEAGENT also yields an almost perfect rating for turn-level actuality and informativeness.

Though ReAct is better than plain LLM, both baselines perform much worse than GENIEAGENT with at least a 50% difference for overall success rate. GENIEAGENT achieves 89% execution success rate once the correct model is selected, while the ReAct agent's execution success rate is 41%. In case studies, we observe that plain LLM and ReAct agents tend to hallucinate input arguments (such as generating a random SMILES sequence or small molecule chains), significantly jeopardizing the execution success rate. The turn-level ratings of the two baselines are also significantly worse than GENIEAGENT. Even though both baselines have access to the scientific searching tools described in section 3.2, the factuality performance is still much worse than GENIEAGENT.

Table 2: Drug discovery model orchestration performance for conversations with experts.

| Method | Overall (0-100%) | | Turn-level (Averaged 1-5 ratings) | | |
| --- | --- | --- | --- | --- | --- |
| | Overall SR | Model Selection SR | Factuality | Progressiveness | Informativeness |
| GENIEAGENT | 50 | 60 | 4.6 | 4.3 | 4.8 |

## 7 EXPERT RATINGS BASED ON EXPERT-INITIATED CONVERSATIONS

### 7.1 EVALUATION SETUP

Four experienced biologists conduct 14 sessions of free-form conversations with the GENIEAGENT. They are not aware of the supported models and the scope of the drug discovery capability ecosystem. They also do not have access to the descriptions or details of each model. Besides participating in the conversation, the experts provide the ratings of whether the conversations end with their needs solved and the ratings of each turn. The expert ratings follow the same metric design described in section 5.3.

### 7.2 RESULTS FROM EXPERT EVALUATION

The performance is demonstrated in Table 2. We observe that the overall success rate and model selection rate are lower than the ones generated by the multi-agent evaluation framework because the experts' questions are more open-ended, in which many questions fall out of the capabilities of GENIEAGENT. The turn qualities mostly fall between 4 to 5, indicating the reliability and helpfulness of GENIEAGENT.

## 8 ABLATION STUDIES

Table 3: Ablation study of various agent design choices.

| | Method | Overall SR | Model Selection SR |
| --- | --- | --- | --- |
| 1 | Fixed Workflow | 39 | 68 |
| 2 | No intention-action | 48 | 57 |
| 3 | No index search | 37 | 45 |
| 4 | No specialized agents | 34 | 65 |
| 5 | No hint nodes | 57 | 70 |
| 6 | GENIEAGENT | 64 | 72 |

We study the effectiveness of the proposed technique in section 4 by ablation study. For fixed workflow, we implement a sequential workflow consisting of several steps for executing a model in the specialized agent. For "no index search", we provide all capabilities as tools directly bind to the primary agent.

The results in Table 3 demonstrate the following observations. 1) Using fixed workflow limits flexibility and hurts the model execution performance, which is handled mainly by specialized agents. 2) Removing reference intention-action retrieval hurts the model selection hit rate by 15 points, indicating the importance of the intention-to-action bridging. 3) Both index searching tools and specialized agents are helpful when selecting the capabilities among the large set of available models. 4) The hint nodes are crucial for keeping the execution on the right track since both hallucinated input and divergent execution steps would compromise the execution success rate.

## 9 CONCLUSION

In this paper, we introduced GENIEAGENT, a novel and scalable drug discovery agent designed to address the limitations of current drug discovery tools. By integrating multiple models under a unified natural language interface, GENIEAGENT streamlines the drug discovery process, enabling cross-tool reasoning, automated model orchestration, and personalized scientific assistance. Our evaluation framework, simulating real-world drug discovery conversations, demonstrates the robustness and reliability of GENIEAGENT, with strong performance in both automated and expert-led evaluations. The results of our large-scale study show that GENIEAGENT significantly outperforms baseline agents, achieving high success rates in model selection and execution, and delivering accurate and informative responses at each turn. Despite the challenges posed by open-ended questions, GENIEAGENT consistently provided solutions to drug engineers' needs, demonstrating its potential as a powerful tool for accelerating early-stage drug development.

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

# A   APPENDIX

## A.1   DETAILS OF THE DRUG DISCOVERY MODELS USED IN THE TOOL ECOSYSTEM

We present the name, source, type, description, input and output data of all models used in the tool ecosystem in Table 4.

Table 4: Machine learning models used as tools in GENIEAGENT for drug discovery.

| | Method | Type | Description | Input | Output |
|---|---|---|---|---|---|
| 1 | Property Enhancer (PropEn) for implicitly-guided antibody generation (Tagasovska et al., 2024) | generative | Uses an encoder-decoder approach to optimize any property of an antibody by pairing similar sequences based on the defined criteria | heavy chain, light chain (AHo-numbered), property | heavy chain, light chain, edit distance to initial antibody |
| 2 | Antibody maturation with guided sampling Gruver et al. (2023b) | generative | Samples from multi-task fine-tuned protein language model and uses the antibody-antigen binding predictions for guidance | heavy chain, light chain, target, objective to guide sampling, regions to redesign, max number of edits, hyperparameters | heavy chain, light chain |
| 3 | Small molecule generation with neural empirical Bayes (NEBULA) (Nowara et al., 2024) | generative | Uses a latent 3D generative model for the scalable generation of large molecular libraries around a seed compound of interest. Sampling is performed in the learned latent space of a vector-quantized variational autoencoder | SMILES | SMILES |
| 4 | Developability predictions (InSilicoMA) Park & Izadi (2024) | scoring | Predicts antibody developability by assessing a set of structural and physics-based molecular surface descriptors | heavy chain, light chain | electrostatic potential, accessibility, binding motifs, electrostatic and hydrophobic interactions |
| 5 | Antibody expression and antibody-antigen complex prediction (Gruver et al., 2023a) | scoring | Uses a multi-task fine-tuned protein language model to predict antibody-antigen complex property prediction | heavy chain, light chain, antigen sequence | probability of binding, binding KD, expression probability, expression yield |
| 6 | Therapeutic Antibody Profiler (hydrophobicity & charge descriptors) Raybould & Deane (2022) | scoring | A high-throughput computational developability assessment tool that assesses the physicochemical "druglikeness" of an antibody candidate | heavy chain, light chain | CDR length, patches of surface hydrophobicity, patches of positive charge, patches of negative charge and structural charge symmetry |
| 7 | ABangle Dunbar et al. (2013) | scoring | Calculates the relative orientation between the variable domains orientation for any antibody and compares with all other known structures | heavy chain, light chain | abangle, main descriptor, other metadata |
| 8 | PEP-Patch (electrostatics estimation) Hoerschinger et al. (2023) | scoring | Visualizes and quantifies the electrostatic potential on the protein surface in terms of surface patches, denoting separated areas of the surface with a common physical property | a topology with bonds in PDB file, structure file or trajectory, SMILES string used to assign bond orders to the topology | positive and negative patch |

Table 5: Machine learning models used as tools in GENIEAGENT for drug discovery (continued).

| | Method | Type | Description | Input | Output |
|---|---|---|---|---|---|
| 9 | Hydrophobicity estimation Waibl et al. (2022) | scoring | Computes the Spatial Aggregation Propensity (SAP) score, a predictive measure of protein aggregation based on molecular simulations. Identifies hydrophobic regions prone to aggregation and estimates per-residue contributions to overall hydrophobicity | heavy chain, light chain | SAP score, estimated hydrophobicity of full Fv, per residue decomposition, aggregation-prone region identification |
| 10 | Protein preparation with SPRUCE Baell & Holloway (2010) | scoring | Automates the process of converting experimentally solved or modeled protein and nucleic acid structures into formats suitable for downstream applications like docking or molecular simulations | REC file | OEDesignUnit |
| 11 | Small molecule docking with POSIT Baell & Holloway (2010) | scoring | Performs ligand docking by leveraging known experimental binding modes to guide the placement of small molecules in the receptor binding site, improving accuracy when structural information is available | REC file, SMILES | docked ligand poses, ranked by predicted binding affinity |
| 12 | Small molecule docking with HYBRID Baell & Holloway (2010) | scoring | Utilizes a combination of ligand-based and structure-based docking approaches to predict binding poses, incorporating both receptor shape and chemical similarity to known binders for enhanced accuracy | REC file, SMILES | docked ligand poses, ranked by predicted binding affinity |
| 13 | Scoring poses using HYBRID Baell & Holloway (2010) | scoring | Evaluates docked ligand poses based on a hybrid scoring function that considers receptor-ligand interactions and known ligand similarities, producing affinity estimates for each pose | protein PDB file, ligand PDB file | docking score, ranked ligand poses |
| 14 | GNINA scoring McNutt et al. (2021) | scoring | Uses a deep learning-based scoring function to evaluate docked ligand poses against a receptor, assigning a ranking score to reflect binding affinity | protein PDB file, ligand PDB file | docking score, ranking of ligand-protein interactions |
| 15 | GNINA docking McNutt et al. (2021) | scoring | Performs molecular docking using a deep learning-based scoring function to predict the optimal binding pose of a ligand in a protein's binding site. The method outputs a ranked list of poses based on predicted affinity | REC path and SMILES, scoring function | file containing ligands and pockets, ranked poses |
| 16 | Drug metabolite prediction Coley et al. (2017); Djoumbou-Feunang et al. (2019) | generative | Predicts how a small-molecule drug gets metabolized by the liver and generates structure(s) of drug metabolites and/or sites of metabolism (nodes in the input structure) | SMILES | metabolites, metabolic reaction description, metabolite probability, confidence scores |

## A.2 PROMPTS

### A.2.1 SYSTEM ROLE OF THE PRIMARY AGENT

You are an assistant for scientists working on drug discovery. You need to use the provided tools to find the helpful functions to help the scientist to call the functions and generate new molecule sequences. If a user shows an intention to call a specific model, call the corresponding function directly, do not ask for input needed for that model from the user.

### A.2.2 SYSTEM ROLE FOR THE SPECIALIZED AGENT

You are a specialized assistant for handling the execution of the drug discovery model MODEL_NAME. The primary assistant delegates work to you whenever the user needs help to execute MODEL_NAME.

You will first introduce this model to the user using the information provided by the 'introduce_MODEL_ID' tool. Then you should ask the user to provide input arguments. Do not hallucinate or guess the arguments, the arguments have to be part of the user input. After that, you will need to validate the input arguments on your own with the 'get_input_MODEL_ID' tool. You will then verify the input arguments with the user to obtain their confirmation. After getting confirmation, you can execute the model with 'execute_MODEL_ID' tool.

When you confirm the input arguments, use a markdown table to show the existing arguments.

If you need more information or the user changes their mind, escalate the task back to the principal assistant. Remember that execution isn't completed until after the 'execute_MODEL_ID' tool has successfully been used.

If the user needs help, and none of your tools are appropriate for it, then "CompleteOrEscalate" the dialog to the host assistant. Do not waste the users' time. Do not make up invalid tools or functions.

### A.2.3  Hint Message for Verifying Critical Actions

All inputs are provided for the MODEL_ID model. Remember to request explicit confirmation to make sure all inputs are correct before executing the model!

### A.2.4  Hint Message for Following Action Plan

You just finish the execution of an action MODEL_ID. Please recall that the complete action plan is ACTION_PLAN_FROM_MEMORY. Make sure you continue following the plan and executing the next planned model.

### A.2.5  Hint Message for Handling Hallucinated Values

A new value is saved to memory. If this value is not provided by the user explicitly or the user explicitly asked you to generate this sequence, this value might be hallucinated. Do not include the values in the input if you think they are generated incorrectly.

