# OpenReview forum: "Orchestrating Tool Ecosystem of Drug Discovery with Intention-Aware LLM Agents"
_ICLR.cc/2025/Workshop/AgenticAI — ICLR 2025 Workshop AgenticAI Poster_

### Official Review · Reviewer_1GAi · 2025-03-04
**Measure the agreement between human annotators and the LLM judge on three turn-level quality metrics**

**Rating:** 9
**Confidence:** 4

**Review:**

**Paper Summary**

This paper introduces GenieAgent, an innovative drug discovery agent characterized by three core components. Firstly, it retrieves reference intention-action pairs to guide Large Language Models (LLMs). Secondly, it integrates metadata-indexed searching tools and specialized agents for capability-focused subtasks, allowing for a division between high-level planning and low-level subtask execution. Lastly, it incorporates hint nodes that activate LLMs for predefined follow-up subtasks under specific conditions, thereby achieving a balance between workflow controllability and flexibility. Additionally, the paper proposes an automatic agent evaluation framework, which includes test samples, an evaluation agent, and an LLM judge. Test samples are derived from real-world drug discovery experiments, with user intentions generated by models. The evaluation agent simulates a drug discovery scientist requesting GenieAgent to resolve the intended objectives, while the LLM judge evaluates turn-level intermediate quality across three quality dimensions. These assessments, combined with the overall success rate, serve as metrics for evaluating drug discovery agents. A series of experiments validate the effectiveness of the agent framework and its individual components.

**Strengths**

The design of the GenieAgent system is meticulous, particularly with its hint nodes mechanism, which effectively balances workflow controllability and flexibility. The evaluation framework is impressive, as it mirrors human scientific thinking by considering a spectrum of user intentions from vague to clear. Experiments further substantiate the effectiveness of GenieAgent, demonstrating its state-of-the-art performance.

**Weaknesses**

It would be beneficial to measure the agreement between human annotators and the LLM judge on three turn-level quality metrics using Cohen’s kappa[1] or other methods.

Furthermore, conducting an error analysis of the GENIEAGENT would be beneficial, as it could identify existing challenges and outline directions for future research.



 [1] McHugh, Mary L. "Interrater reliability: the kappa statistic." Biochemia medica 22.3 (2012): 276-282.

---

### Official Review · Reviewer_5YHq · 2025-03-05

**Rating:** 6
**Confidence:** 4

**Review:**

The paper introduces GENIEAGENT, an intention-aware AI agent designed to orchestrate drug discovery tools using Large Language Models (LLMs). The drug discovery process relies on various computational models for molecular design, scoring, and ranking, but existing approaches lack seamless integration and adaptability to user intentions. GENIEAGENT addresses these issues by providing a unified natural language interface that bridges scientific intentions to concrete tool usage, enabling cross-tool reasoning and dynamic workflow orchestration. The system incorporates specialized agents, indexed search utilities, and hint nodes to balance structured guidance with open-ended exploration. A novel evaluation framework simulates scientific discovery conversations, and results show that GENIEAGENT significantly outperforms baseline AI agents, achieving higher scientific accuracy and robustness when assisting molecular engineers.

Suggested Improvements:
1. The authors can introduce domain-agnostic agent modules to improve generalization beyond drug design.
2. The authors can add explainability features for scientific decision-making, allowing users to audit the AI’s reasoning process.

---

### Decision · Program_Chairs · 2025-03-05

Accept (Poster)